# Yoked surface codes

Craig Gidney[1,3], Michael Newman [1,3] ✉, Peter Brooks[2] & Cody Jones[1]

One of the biggest obstacles to building a large scale quantum computer is the high qubit cost of protecting quantum information. For two-dimensional architectures, the surface code has long been the leading candidate quantum memory, but can require upwards of a thousand physical qubits per logical qubit to reach algorithmically-relevant logical error rates. In this work, we introduce a hierarchical memory formed from surface codes concatenated into high-density parity check codes. These yoked surface codes are arrayed in a rectangular grid, with parity checks (yokes) measured along each row, and optionally along each column, using lattice surgery. Our construction assumes no additional connectivity beyond a nearest-neighbor square qubit grid operating at a physical error rate of $10^{-3}$. At algorithmically-relevant logical error rates, yoked surface codes use as few as one-third the number of physical qubits per logical qubit as standard surface codes, enabling moderate-overhead fault-tolerant quantum memories in two dimensions.

The surface code is a leading quantum error correcting code for building large scale fault-tolerant quantum computers because of its forgiving qubit quality and connectivity requirements[1]. The surface code's major downside is its extremely demanding quantity requirements. At an error rate of $10^{-3}$, it can take 1000–2000 physical qubits per logical qubit for the surface code to reach error rates low enough to run classically intractable algorithms[1–5].

There are many ideas in the field for reducing this overhead[6–19], as well as bounds on possible improvements[20–24]. Constructions for reducing overhead frequently require high-fidelity long-range connections, which can be difficult to engineer in architectures like superconducting qubits. When restricting to nearest neighbor planar connectivity, one strategy is to concatenate the surface code into an outer code with a higher ratio of logical qubits to physical qubits[25]. The surface code provides high quality qubits, which the outer code can densely encode with increased protection, hopefully using fewer qubits than simply expanding the surface codes directly. The surface code also provides mechanisms like lattice surgery[26,27] to perform operations between distant qubits.

We usually imagine that the overlying code should have a high code distance, a high code rate, and low-density parity checks (qLDPC). Small parity checks provide two important advantages. First, their syndrome extraction circuits are small and highly parallelizable[6], so the noise injected into the system while measuring checks is low. Second, their locality limits the damage caused by correlated

errors, sometimes for free[28]. However, some of these properties are less important when using low-error surface code qubits at the base level.

In this work, contrasting with qLDPC-based approaches, we consider simple high-density parity check codes as outer codes. We focus solely on achieving a high coding rate and simple layout, see Fig. 1, and refer to these outer parity checks as yokes. In 1D, these consist of parity checks along each row of an array of surface codes[29]. In 2D, they consist of parity checks along each row and column of the array, up to qubit permutations. These outer codes have distances two and four respectively which, by utilizing the soft information provided by the inner surface codes[30], double and quadruple the inner code distance. The inner surface codes suppress error rates to levels that allow us to measure high-weight checks without incurring significant noise. Furthermore, we can avoid damaging correlated errors introduced when measuring high-weight checks by adding protection against them using a spacetime tradeoff during lattice surgery. We find that these constructions can use as few as one-third as many physical qubits as standard surface codes under the same connectivity constraints. In addition, Supplementary Note 1 contains overhead estimates using standard circuit-level depolarizing noise (as opposed to the noisier superconducting-inspired model employed in the main text), specifications of the noise models, an alternative outer code construction using a single $Y$-type check, and higher-dimensional generalizations of quantum parity check codes.

[1]Google Quantum AI, Santa Barbara, CA, USA. [2]Google, Sunnyvale, CA, USA. [3]These authors contributed equally: Craig Gidney, Michael Newman.
✉e-mail: mgnewman@google.com

**standard (unyoked) surface codes**

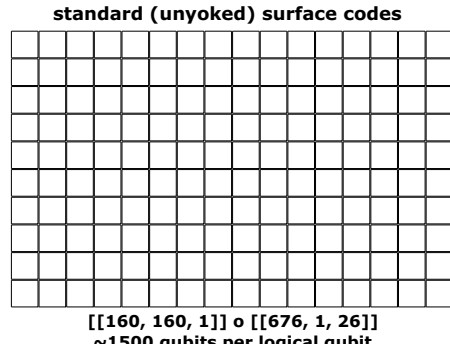

[[160, 160, 1]] o [[676, 1, 26]]
~1500 qubits per logical qubit

**1D yoked surface codes**

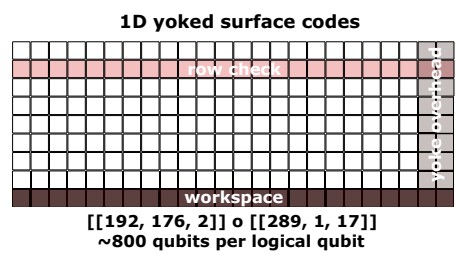

[[192, 176, 2]] o [[289, 1, 17]]
~800 qubits per logical qubit

**2D yoked surface codes**

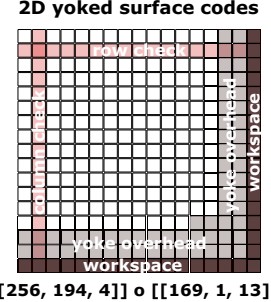

[[256, 194, 4]] o [[169, 1, 13]]
~600 qubits per logical qubit

**Fig. 1 | Yoked surface code footprints.** From left to right: unyoked, 1D, and 2D yoked surface code patches drawn to relative scale. In each row of 1D yoked surface codes, we measure multi-body logical *X*- and *Z*-type stabilizers. In 2D yoked surface codes, we additionally measure multi-body logical *X*- and *Z*-type stabilizers in each column. The *Z*-type stabilizers are applied to a permutation of the 2D code to commute with the *X*-type stabilizers. Gray patches represent overhead introduced by the outer stabilizers (i.e. yokes). Dark patches represent the workspace required to measure the row/column stabilizers. There is also overhead due to interstitial space between patches for lattice surgery. Concatenated code parameters, along with approximate overall qubit footprints (including the various overheads) labeled below. Note that the [[192, 176, 2]] outer code is a collection of eight [[24, 22, 2]] 1D parity check code blocks. The logical qubits can be reliably stored for about a trillion operations assuming a physical error rate of $10^{-3}$ in a superconducting-inspired noise model. The relative savings of yoked surface codes over unyoked surface codes grows as the target error rate decreases.

## Results

### Quantum parity check codes

Quantum parity check codes are CSS codes that generalize classical parity check (or array) codes. In classical parity check codes, bits are laid out in a 1D or 2D array, and the parity of each row or each row and column are checked. We focus on these codes because their parity checks are geometrically simple, and their rate quickly tends to one as their code block size increases.

Unlike classical parity check codes, we must enforce certain block size restrictions to ensure that the stabilizers of quantum parity check codes commute. In 1D, we require the length of the code to be even. In 2D, we require the side length $n$ of the block to be divisible by four. The reason is that row and column operators of opposite Pauli type anti-commute, since they intersect in a single location. When $4|n$, this can be fixed by permuting the qubits for different Pauli type row and column checks. Let $e_i \in \mathbb{F}_2^{n/2} \times \mathbb{F}_2^{n/2}$ and $f_j \in \mathbb{F}_2^2 \times \mathbb{F}_2^2$, where $(e_i)_{kl} = \delta_{ik}$ and $(f_j)_{kl} = \delta_{jk}$. Then we can modify the support of the *Z*-type row and column parity checks of an $n \times n$ 2D parity check code as

$$e_i \otimes f_j \mapsto e_i \otimes f_j^\top \tag{1}$$

$$e_i^\top \otimes f_j^\top \mapsto e_i^\top \otimes f_j \tag{2}$$

to ensure that they commute with row and column *X*-type checks. As an example, we describe the stabilizers and observables of the [[64, 34, 4]] 2D parity check code in Table 1.

More generally, we require an $r$-dimensional parity check code to have each side length divisible by $2^r$, see Supplementary Note 4. An $r$-dimensional parity check code has distance $2^r$, with minimum weight logical operators forming the vertices of $r$-dimensional rectangles. We can count the number of independent checks to determine it has parameters $[[\prod n_i, 2\prod(n_i - 1) - \prod n_i, 2^r]]$, where $n_i$ are the side lengths of the $r$-dimensional array. In particular, for 1D and square 2D codes, we obtain families of $[[n, n - 2, 2]]$[29] and $[[n^2, n^2 - 4n + 2, 4]]$ codes, respectively.

### Lattice surgery constructions

A key detail when concatenating a code over the surface code is how the checks of the overlying code will be measured. This is important as the workspace needed to periodically measure the checks of the overlying code can be even larger than the space needed to just store the qubits. Furthermore, we must account for the effects of errors occurring during the lattice surgery in order to ensure fault-tolerance of the outer code.

Originally, lattice surgery was thought of as being built out of parity measurements[26,31], but another useful perspective is to view lattice surgery as an instantiation of the ZX calculus[32]. In the latter perspective, the building blocks of lattice surgery are not operations on qubits but rather connections between junctions. Making a good lattice surgery construction then becomes an exercise in packing, routing, and rotating pipes so that they link together in the required way. We include an example connecting topological diagrams to the corresponding lattice surgery steps in Fig. 2.

In Fig. 3 we illustrate example multi-qubit measurements performed using lattice surgery. Typically, when executing a fault-tolerant circuit described in terms of these topological diagrams, we don't concern ourselves with which error strings occur. We assume that any error string will corrupt the circuit. However, when concatenating the surface code into an outer code, we must worry about error propagation in much the same way we do when designing fault-tolerant syndrome extraction at the base code level. One disadvantage of using high-density parity check codes is that measuring these high-weight stabilizers can induce correlated failures among the inner surface codes, which might not be corrected by the outer code. These are analogous to hook errors propagating from a measure qubit to many data qubits.

However, unlike physical qubits, surface codes can modulate the distances between boundaries to bias protection against different error mechanisms. Of course, this might enlarge the footprint of the concatenated code by extending the size of the base code. However, a second nice property of the surface code is that its topological operations are mostly agnostic to their spacetime orientation. Consequently, we can orient this extended protection in the time direction, holding the spatial footprint of the yoked surface codes fixed. For example, in Fig. 3, we orient the correlated error in time and extend the protection against it to maintain the concatenated codes effective doubled distance. This isn't a free lunch: it increases the length of the outer code's syndrome cycle, which in turn increases the distance required by the inner code. However, as the error rate scales polynomially with the length of the syndrome cycle and inverse exponentially with the distance of the inner code, this tradeoff is not too damaging.

As 1D and 2D yoked surface codes have doubled and quadrupled code distances respectively, we elect to extend the duration of the yoke checks to ensure each correlated error is protected to distance

**Table 1 | Stabilizers and observables of the [[64, 34, 4]] 2D parity check code**

| [[64, 34, 4]] 2D parity check code | | | | | |
|---|---|---|---|---|---|
| X col checks | X row checks | Z bi-col checks | Z bi-row Checks | Y observables (positioned by location of Y term) | |

**Table 1 (continued) | Stabilizers and observables of the [[64, 34, 4]] 2D parity check code**

### [[64, 34, 4]] 2D parity check code

| X col checks | X row checks | Z bi-col checks | Z bi-row Checks | Y observables (positioned by location of Y term) |
|---|---|---|---|---|

Each cell shows the entries of a stabilizer or observable as an 8 × 8 grid of Pauli terms; one term for each of the 64 physical qubits. Note that both the X observables and Z observables can be recovered from the Y observables, as the X observables contain no Y or Z terms and the Z observables contain no X or Y terms. Also note that two of the listed checks are redundant - the product of all X col checks is equal to the product of all X row checks, and the product of all Z bi-col checks is equal to the product of all Z bi-row checks.

$2d_{inner}$ (as in Fig. 3) and $4d_{inner}$, respectively. This is likely overly conservatives, as it effectively suppresses the 1D and 2D correlated failures using $2d_{inner}$ and $4d_{inner}$ (asymmetric) surface codes, which we will see provide significantly better per-distance protection than yoked surface codes on uncorrelated failures. This makes the probability of a correlated failure negligible relative to the dominant uncorrelated failure mechanism. We could likely shorten the syndrome cycle considerably by using a workspace whose spacelike and timelike distances are merely the distance of an unyoked surface code giving a similar error per patch-round. This may be sufficient to suppress the correlated (and with increased workspace size, measurement) error probability below the uncorrelated failure probability. However, we err on the side of caution in our estimates, and use overly protected syndrome measurements. Note that this conservative strategy does provide more opportunities for spacelike failure paths, which corresponds to increasing the effective measurement error probability of the outer code.

To construct the full syndrome extraction circuit, we build it up from pieces. In Fig. 4, we show the *X*- and *Z*-type outer stabilizers measured by combining parity measurements with patch rotations from ref. 27. Finally, we fit these puzzle pieces together to form the full syndrome cycle circuits in Fig. 5. Note that there is extra workspace required to measure these checks. For example, in 1D yoked surface codes, we use a single workspace row to sequentially measure several 1D yoked surface code blocks, analogous to a measure qubit migrating across the code blocks to extract stabilizer measurements. Having a single extra row attend to all the blocks lengthens the outer code's syndrome extraction cycle, but reduces the number of occupied surface code patches. Striking a balance between these effects is important, as we must include the overhead of this workspace in the yoked surface code's overall footprint.

We also consider a second storage format, which we call hot storage. In our previous format cold storage, logical qubits were stored as densely as possible, but could not be immediately operated upon. Operating on a logical qubit in cold storage requires first getting it out of storage. Concretely, this means the surface code patches don't all have access hallways available next to them. It's still necessary for some workspace to be present, because it's necessary to periodically check the yokes, but this workspace may be shared between many groups of patches.

Logical qubits in hot storage are available to be operated upon. Each surface code patch has a boundary exposed to an access hallway that provides a route out of storage. The existence of this access hallway is useful for yoked surface codes because it can also be used as a workspace for periodically measuring the yokes. This means measuring the yokes has no marginal space cost; the space was already paid for. Instead, it has a marginal spacetime cost, because although the access hallway was already there, it's blocked while a yoke is being measured. Figure 6 shows the space layout, and spacetime layout, that we use for estimating the overhead of hot storage.

In this work, we only consider hot storage of 1D yoked surface codes. We expect the access hallway requirements and access hallway utilization of 2D yoked surface codes would be more demanding. In fact, storage could be even hotter than we consider here, by having each surface code patch expose two boundaries[5,27]. However, we expect the additional space cost (naively at least 1.5×) of exposing two boundaries would likely mitigate any benefit of using 2D yoked surface codes or the computational benefit of having both bases immediately available. Consequently, qubits in hot storage are rotated on an as-needed basis when accessing different boundaries. The need to perform these rotations is a key consideration when laying out an algorithm.

Yoked surface codes in hot storage can be operated on by lattice surgery while they're encoded. Each encoded observable is spread over several surface code patches, but lattice surgery can stitch

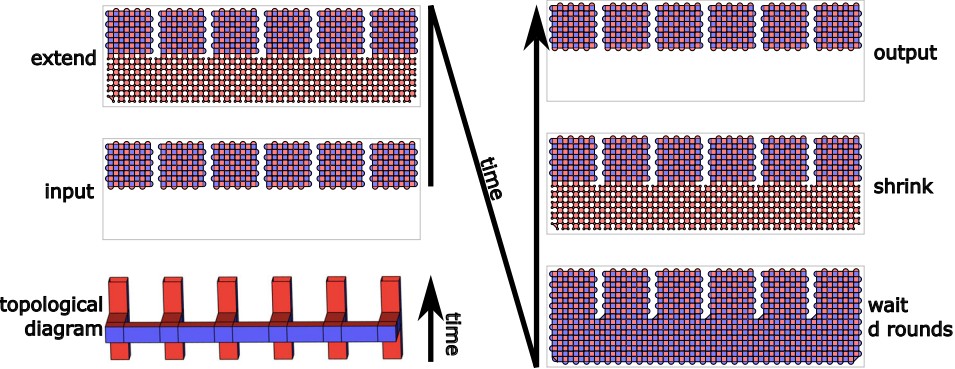

**Fig. 2 | Connecting topological diagrams to lattice surgery.** Every topological diagram is a 3D representation of surface code lattice surgery evolving in time. Each block has spacetime extent $d \times d \times d$ for surface code distance $d$. Extended connections between blocks serve as a visual aid to emphasize the topology. This shows the topological diagram for a multi-body $Z$-measurement, accompanied by the stabilizers measured in each time slice.

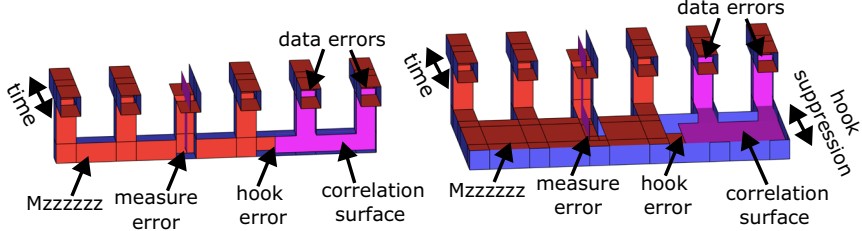

**Fig. 3 | Topological diagrams of multi-body logical measurement.** Left: a multi-body $Z$-measurement, with some surfaces removed for clarity. The right-hand correlation surface shows the equivalence of a short spacelike hook error to two data errors. The left-hand correlation surface shows a measurement error, which is equivalent to two data errors immediately before and after the measurement. Right: the same multi-body $Z$-measurement with protection against correlated hook errors. We can increase protection against the hook error by extending the distance between the boundaries that it connects. Naively, this would increase the overall qubit footprint of the circuit. However, we can orient this extension in time, trading a smaller qubit footprint for a longer syndrome extraction cycle. This also reorients the measurement error to be spacelike.

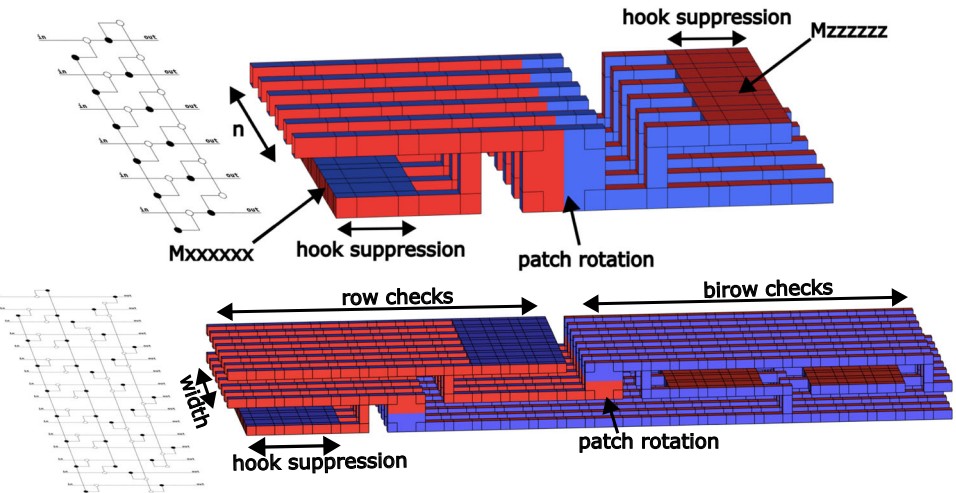

**Fig. 4 | Checking yoke stabilizers using lattice surgery.** Time flows left to right, with the corresponding ZX diagrams shown on the left. Top: For 1D yoked surfce codes, the process occupies $2n$ surface code patches for $8d$ rounds, where $n$ is the block length of the outer code. Bottom: For 2D yoked surface codes, the process occupies $3w$ surface code patches for $25d$ rounds, where $w = \sqrt{n}$ is the the width of the outer array. In the ZX diagram, top pipes correspond to every other wire beginning from the top, while bottom pipes correspond to every other wire beginning second from the top.

together several patches as easily as one. The cost is essentially identical to doing lattice surgery with unyoked surface codes, because the entrance to the access hallway is occupied regardless of how many patches are touched. However, we must be careful when operating on these encoded qubits and ensure that we do not frequently expose unprotected lower-distance patches. Consequently, to keep our overhead estimates conservative, we assume access hallways require the full unyoked code distance in height.

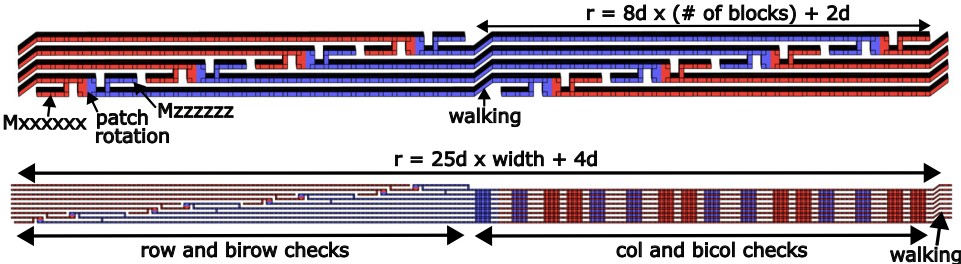

**Fig. 5 | The full syndrome extraction of yoked surface codes.** Top: For 1D yoked surface codes, an extra row of workspace surface code patches travels through the blocks to measure the outer stabilizers. We use the walking surface code construction in ref. 40 to connect the outer syndrome cycles, which takes $2d$ rounds to execute. The total length of a syndrome cycle scales as $8d \times$ (# of blocks) $+ 2d$. Bottom: For 2D yoked surface codes, an extra row and column of workspace surface code patches travels through the blocks to measure the outer stabilizers. We use two iterations of walking surface codes to connect the outer syndrome cycles, which together take $4d$ rounds to execute. The total length of a syndrome cycle scales as $25dw + 4d$, where $w = \sqrt{n}$ is the width of the square outer code array, in this case 8.

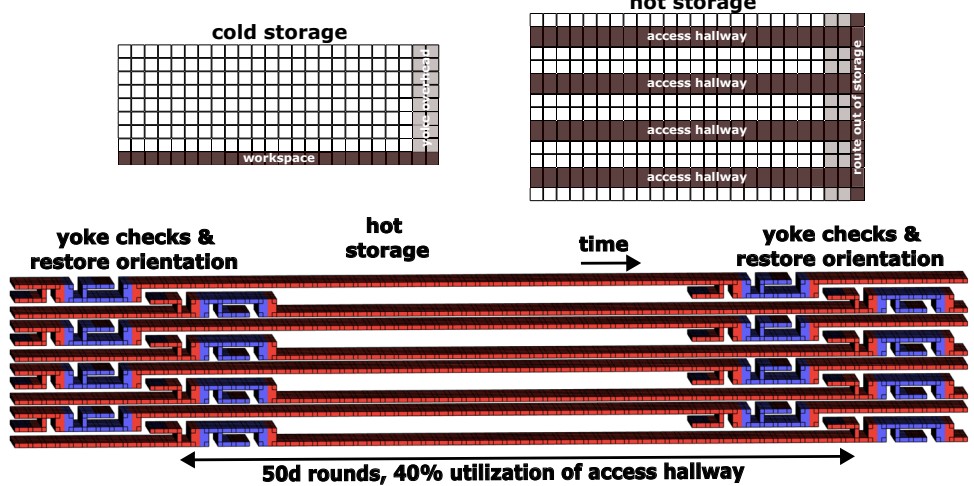

**Fig. 6 | Layouts for cold and hot storage using 1D yoked surface codes.** The 2D footprint diagrams at the top show how space is allocated, while the 3D topological diagrams at the bottom show hot storage syndrome cycles occurring over time. In the 2D footprint diagrams, each row is a separate outer code block. The white-filled squares correspond to usable storage while other squares correspond to various overheads. In cold storage, one row of workspace is shared between different blocks in order to measure the yokes. In hot storage, the yokes are measured using the access hallways that are already present. We assume that the access hallways are the full unyoked code distance in height. The hot storage syndrome cycle shown takes $50d$ rounds and utilizes the access hallways 40% of the time.

Evidently, using yoked surface codes in fault-tolerant computation has consequences on the large scale architecture of a quantum computer, beyond just the size of the storage. Operating on yoked surface codes is important to understand, but beyond the scope of this paper.

**Complementary gaps**

From the perspective of the outer code, the syndrome of the inner code gives valuable information about the likelihood of an error in a particular location. For example, an inner surface code with no detection events is far less likely to have experienced an error than an inner surface code with many detection events. We can quantify this likelihood by comparing the probability of a set of errors obtained from minimum-weight matching against the probability of a set of errors obtained from a minimum-weight matching conditioned on the complementary logical outcome. Passing this information to the outer code helps it to identify likely culprit errors[30].

Operationally, we can compute the minimum-weight matching conditioned on the complementary logical outcome by modifying the error graph. Given a block of surface code memory with boundaries, we can form a detector connecting to all the boundary edges on one side of the error graph, similar to[33,34]. This augmentation maintains the

graph structure, and turning this boundary detector on/off forces the decoder to match/not match to the corresponding boundary. The resulting two matchings are the decoder's best hypotheses for the set of errors explaining these two topologically distinct classes of errors. We call the log-likelihood ratio of these two hypotheses the complementary gap - the log-ratio of the probabilities of the minimum-weight matching and the complementary matching. A complementary gap close to zero indicates that the decoder is not confident in its decision, while a high complementary gap indicates the decoder is highly confident.

This information is extremely helpful and can be used in decoding the outer code. 1D and 2D parity check codes can themselves be decoded using minimum-weight perfect matching, and so we use these complementary gaps as edge weights in the outer error graph. These edge weights represent the cost of flipping a minimum-weight matching to a complementary matching. The complementary gap is converted to a probability which will serve as the effective error rate of the edge of the outer error graph it corresponds to. Note that in practice one must XOR the syndrome produced by the minimum-weight perfect matching into the observed syndrome, as choosing the outer edge represents the outer decoder selecting the complementary matching in place of the minimum-weight matching. Relative to the

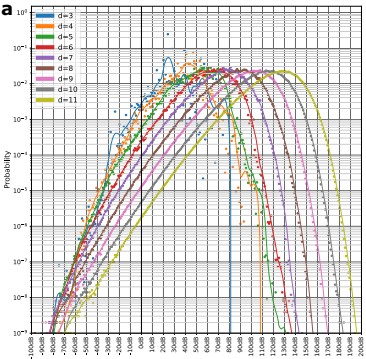
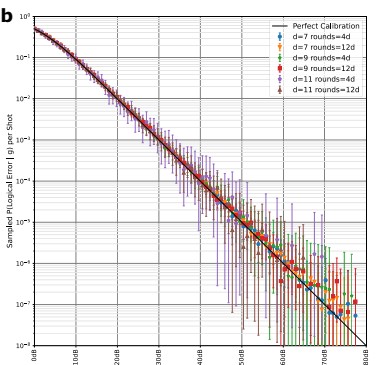
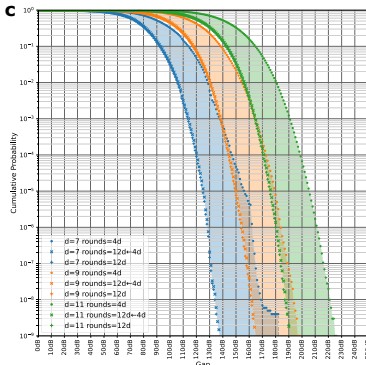

**Fig. 7 | Complementary gap statistics.** We use the SI1000 error model described in Supplementary Note 2 at an error rate of $10^{-3}$. **a** $10d$ round memory experiments with perfect terminal time boundaries checking one observable. Gaps are presented in terms of their ratio in dB, where each gap is binned into the nearest integer dB. A negative gap indicates that the more likely outcome was incorrect. Overall, each curve is comprised of $10^9$ samples. **b** The decoder calibration after rescaling the gap by 0.9×. Error bars represent hypothetical logical error rates with a Bayes factor of at most 1000 versus the maximum likelihood hypothesis probability, assuming a binomial distribution. **c** Extrapolating the inverse cumulative distribution functions of the complementary gaps by exponentiating. Extrapolations are represented as x's with the intervening space shaded. This well-approximates the inverse cumulative distribution function of a longer memory experiment, with a slight tendency towards sampling too-large gaps as the gap increases.

usual decoding problem, the outer decoder's prior changes shot-by-shot depending on the configuration of detection events in the inner codes. While we have selected matchable outer codes, this procedure easily generalizes to setting the prior of any outer decoding problem (possibly replacing edge probabilities with hyperedge probabilities) so long as one has a suitable outer code decoder.

There are several ways to generalize this procedure to a correlated matching decoder. In this work, we use a two-pass correlated matching decoder similar to the one described in ref. 35. To compute the complementary gap in a $Z$-basis memory experiment, we use the $X$-type error graph to reweight the $Z$-type error graph, and then compute the complementary gap for this reweighted $Z$-type error graph.

In Fig. 7, we see that the gap distributions take a smooth, simple form after an initially noisy start at low distance, likely due to finite-size effects. For our benchmarks, it will be important to extrapolate the behavior of these gaps, as we will use them to estimate the behavior of very large simulations at low error rates. We observe that the gaps are well-calibrated - the likelihood of success predicted by the gap is close to the true empirical likelihood of success after rescaling the gap by 0.9×. That is, we rescale the decoder's confidence to account for its slight over-confidence in high-confidence predictions. The decoder also remains well-calibrated when extended over many rounds. We also observe the distribution on complementary gaps over $mn$ rounds can be well-approximated as the minimum of $m$ samples from the distribution on gaps over $n$ rounds. Although these approximations tend to be slightly optimistic, they allow us to extrapolate the probability of observing a particular gap and the resulting likelihood of failure from a distribution of gaps on relatively few (e.g. a small multiple of $d$) rounds.

**Benchmarking**

Simulating yoked surface codes introduces some difficulties. We want to probe very low error rates on blocks of very many logical qubits. Consequently, we perform two types of simulations.

The first type of simulation is a circuit-level simulation of the inner code over a relatively small number of inner rounds, with a single perfect outer round of yoke checks. Yoke detectors are placed along the boundaries of the inner surface code error graphs, with yoke detectors occupying either one or two boundaries of the error graph in the case of 1D or 2D outer codes. This turns decoding of the concatenated code into a single minimum-weight matching problem. Note that, when casting concatenated decoding as a single matching problem, the yoke detectors must connect to the boundary. Otherwise, individual errors might introduce more than two detection events on a particular error graph.

For longer simulations, we phenomenologically simulate multiple rounds of the outer codes using gap distributions. The effective error rate seen by the outer code is determined by the distance of the inner code, as well as the approximate spacetime volume that contributes to a particular error edge. We consider spacelike and timelike edges only. While some of the spacelike error probability may be redistributed to spacetime-like edges in 2D outer codes (depending on different lattice surgery schedules), we do not expect this added complexity to introduce a significant effect. In fact, we will see that timelike edges do not contribute significantly to the overall error rate, which appears to be dominated by shortest-path error configurations. We also ignore damaging hook errors, having already paid to suppress them below the relevant noise floor by using the hook suppressed lattice surgery construction.

These simulations proceed as follows. First, we simulate the complementary gap distribution over $10^9$ shots and varying code distances at $10d$ rounds of a $Z$-type memory experiment - these distributions (binned by nearest integer dB) are recorded in Fig. 7. We record the probability of observing a particular gap as well as the likelihood that particular gap results in an error, i.e. that the minimum-weight matching does not belong to the same error class as the true error configuration. Frequently, we might sample gaps for which we have seen no failures. Extrapolating from the calibration curve in Fig. 7, we assign these events a likelihood of failure corresponding to 0.9× the sampled gap. For each edge in the outer error graph, we associate a number of rounds $N$ that contribute to an error resulting in flipping that edge. Heuristically, we focus on the time extent between rounds to determine $N$. Some components (like patch rotations) can have increased spatial extent, but also frequently come with relatively fewer shortest paths or average out against increased protection during a later time step. We treat timelike outer edges separately, counting rounds according to their spacetime extent, since they extend much farther in space than time. To extrapolate the complementary gap distributions, we raise the inverse cumulative distribution function to the $\frac{N}{10d}$ power, corresponding to taking the minimum gap over $\frac{N}{10d}$ draws from the $10d$ round complementary gap distribution.

Finally, we perform our simulations by sampling, for each edge in the isomorphic $Z$- and $X$-type error graphs, the complementary gap and the corresponding likelihood of failure. In 1D yoked surface codes, these are timelike line graphs with multiple boundary edges at each node. In 2D yoked surface codes, these are layers of complete bipartite

graphs between row check detectors and column check detectors, with timelike edges connecting the layers. We sample gaps to determine which edges have suffered an error, and use those gaps to weight the edges. Finally, we decode these weighted outer error graphs using minimum-weight matching. Note that separately simulating and decoding the Z- and X-type error graphs could negate useful correlations between them, but these benefits are likely small given the surface code's bias against logical Y-type errors.

Following Fig. 7, these simulations use the SI1000 noise model specified in Supplementary Note 2. There, we also include benchmarks using a standard uniform depolarizing model.

### Scaling approximations

In order to flexibly choose the inner code distance, code block size, and number of code blocks to meet a target logical error rate, our simulations validate simple, heuristic scaling laws that we can use to estimate the logical error per patch-round. This quantity can in turn be used to estimate the overall logical error rate given the lattice surgery constructions so that we can choose the most compact layout that realizes said target.

Normal surface code patches have a logical error probability that scales inverse exponentially in the patch diameter ($d$), linearly versus round count ($r$), and linearly versus patch count ($n$). Consequently, the asymptotic scaling of normal patches is $\Theta(r \cdot n \cdot \lambda_0^{-d})$ for some error suppression factor $\lambda_0$. At lower error rates, a simple way to predict this scaling is to focus on the length and number of shortest error paths. Increasing the patch diameter increases the shortest path length, which is the source of the exponential suppression versus $d$. Doubling the number of rounds (or patches) doubles the number of shortest paths, which causes linear scaling in $r$ (or $n$).

Yoked surface code patches change the length and number of shortest error paths. In 1D yoked surface codes, shortest error paths correspond to two shortest error paths in different surface code patches. This doubles the code distance, but introduces quadratic scaling in $r$, the number of rounds between yoke checks. Furthermore, since a logical failure can occur over any two patches in the code, the number of shortest error paths also scales quadratically in $n$. Based on this, we expect the scaling of the 1D yoked logical error rate to be $\Theta(r^2 \cdot n^2 \cdot \lambda_1^{-d})$, where $\lambda_1$ denotes the increased error suppression factor obtained from doubling the code distance, which we would expect to be at most $\lambda_0^2$. Note that the quadratic scaling versus $r$ is only for numbers of rounds up to the yoke check period. Shortest paths from patches in different check periods don't combine to form a shortest error path in the concatenated code (although they can form an effective measurement error). Consequently, we expect the scaling versus rounds to be quadratic up to the check period, and then linear afterwards.

A similar argument holds for 2D yoked surface codes, for which the distance quadruples, but now the error scaling becomes quartic versus the number of rounds $r$ between yoke checks, while remaining quadratic in the number of patches $n$. The quartic scaling versus $r$ is similar: within the support of a weight four logical error, any combination of error paths within those patches can cause the error. The quadratic scaling versus $n$ is because weight four logical errors correspond to four logical errors landing on the corners of a rectangle within the grid. There are $\Theta(n^2)$ of these rectangles specified by their endpoints across a diagonal. Based on this, we expect the scaling of the 2D yoked logical error rate to be $\Theta(r^4 \cdot n^2 \cdot \lambda_2^{-d})$, where $\lambda_2$ denotes the increased error suppression factor obtained from quadrupling the code distance, which we would expect to be at most $\lambda_0^4$.

We emphasize that these are simple path-counting heuristics, where we can empirically fit the different $\lambda$ factors and coefficients. However, these can of course break down. For example in 2D, the set of higher-weight inner logical errors leading to failure could scale as $o(r^4)$.

Despite this, we observe good agreement between simulation and these heuristic scaling laws.

### Numerics

We first validate the gap simulation against the full single-outer-round circuit simulation of the inner surface codes - see Fig. 8. The full simulations were performed by generating stim circuits[36] to describe the experiments, with yoke detectors added to the boundaries to preserve matchability. We then decode these experiments using correlated minimum-weight perfect matching. The circuits that we generate for both generating gap distributions and performing the full simulations use the gateset $\{U_1, CZ, M_Z, R_Z\}$ and the superconducting-inspired noise model SI1000 specified in Supplementary Note 2 at an error rate of $10^{-3}$. Estimates with more standard uniform circuit-level depolarizing noise are also included in Supplementary Note 1. We use noiseless time boundaries where the stabilizers and observables are prepared or measured noiselessly. It's also important that the circuits have many rounds to reduce distortions from the noiseless time boundaries as well as minimize the imperfect extrapolation of the gap distribution. All circuits that we run use gaps extrapolated from $10d$ round distributions, where $d$ is the inner surface code patch diameter.

Normally, we would simulate more varied error rates. The underlying reason for focusing on one error rate is that Monte Carlo simulation of even small yoked surface codes is rather expensive, as the natural size scales span orders of magnitude more qubits and rounds than normal memory experiments. We were particularly interested in understanding scaling with respect to size, rather than with respect to noise strength, so we sacrificed noise strength diversity in favor of size diversity.

We observe good agreement between the gap simulation and full simulation in 0D and 1D, as well as fairly good agreement in 2D. We suspect the small deviations are a result of imperfections in the extrapolation of the gap distribution magnified by the distance-4 outer code. From these points, we can establish single-significant-figure fits that well-approximate these error rates and are consistent with the path-counting scaling approximations. These fits are

$$p_{L,0} \approx r_i \cdot n \cdot 3^{-d}/20 \tag{3}$$

$$p_{L,1} \approx r_o \cdot r_i^2 \cdot n^2 \cdot 8^{-d}/500 \tag{4}$$

$$p_{L,2} \approx r_o \cdot r_i^4 \cdot n^2 \cdot 50^{-d}/200000, \tag{5}$$

where $p_{L,k}$ indicates the cumulative logical error rate of $k$-D yoked surface codes over $r_o$ rounds of the outer code, with $r_i$ rounds between checks, for size-$n$ code blocks, and distance-$d$ inner surface codes. Note that, relative to the $3^{-d}$ error suppression factor of the standard surface code, the error suppression factors in 1D (2D) of $8^{-d}$ ($50^{-d}$) fall short of the optimal $9^{-d}$ ($81^{-d}$) projected from doubling (quadrupling) the code distance.

Our final simulations are phenomenological simulations over many rounds. In particular, we simulate sizes on the order of the maximum extents of our extrapolations, up to $200d$ rounds between checks and outer blocks of 256 surface code patches. We simulate 10 rounds of the outer code, with a fixed 100 $d \times d \times d$ blocks of spacetime contributing to measurement error, which is greater than any of our constructions (see Fig. 4). In the limit of very many $d \times d \times d$ blocks contributing to measurement error, the system should behave like a one-round experiment with $r_o \cdot r_i$ inner rounds as the measurement edges approach weight zero, up to identifying degenerate edges. However, for the sizes we consider, it appears that the (spacelike) minimum-weight error paths dominate the scaling behavior.

We present gap simulations of these long phenomenological benchmarks in Fig. 8. We observe excellent agreement in 0D with the

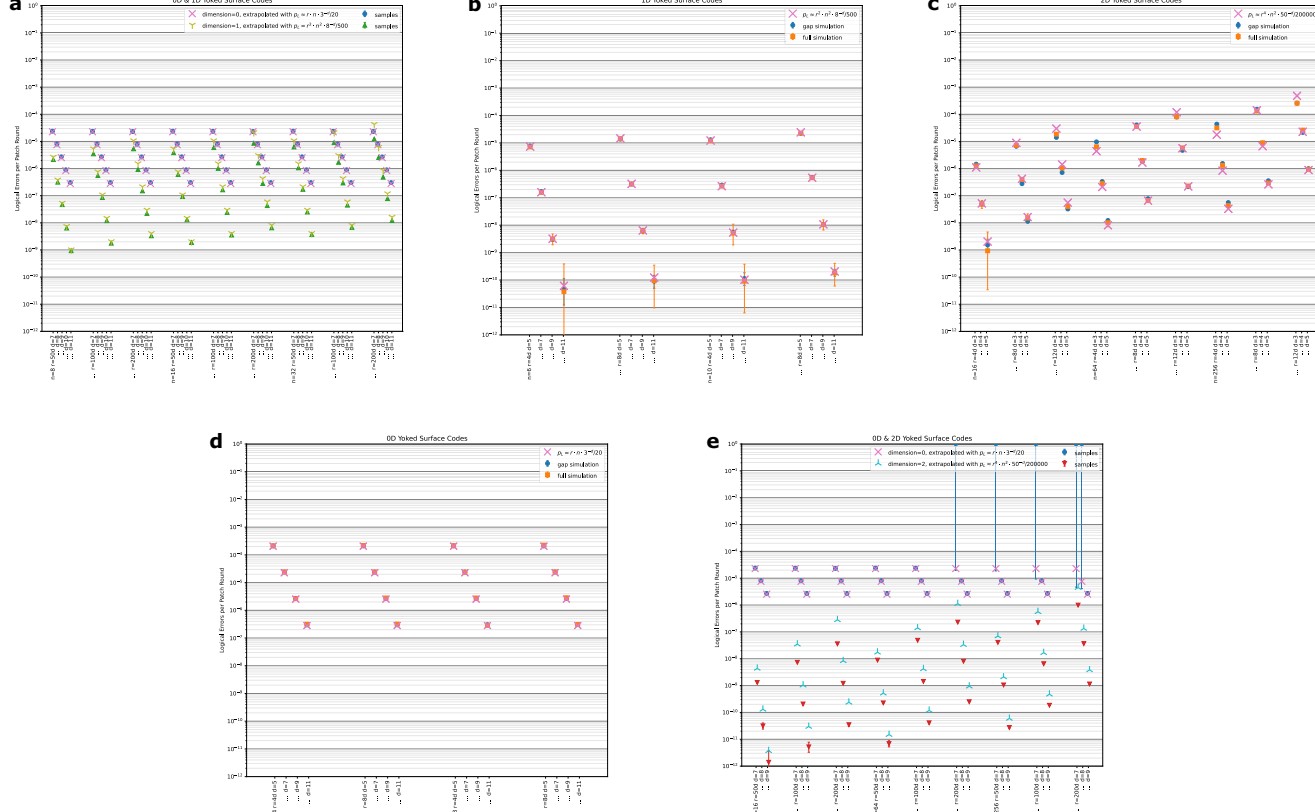

**Fig. 8 | Simulations of yoked surface codes.** We perform simulations over different code block sizes $n$ and different numbers of rounds $r$ in the check. We include single-significant-figure fits consistent with the different path-counting scalings that well-approximate the data. **a**–**c** A comparison of full simulation versus gap simulation. The error rates are reported in terms of logical error per patch-round. **d**, **e** Phenomenological gap simulations of yoked surface codes. Each trial consists of 10 outer code rounds, with a fixed measurement error rate given by a 100$d$ round gap distribution. These simulations assume the hook error has been suppressed below the noise floor set by other error mechanisms. The surface code data points toward the very top right correspond to highly noisy simulations, but with error bars that cover the extrapolations. Error bars represent hypothetical logical error rates with a Bayes factor of at most 1000 versus the maximum likelihood hypothesis probability, assuming a binomial distribution.

scaling approximations. For 1D yoked surface codes, we see good agreement, with a slight tendency for the scaling approximations to predict too high a logical error rate. For 2D yoked surface codes, we see a significant deviation from the scaling approximations predicted from smaller experiments. There are several potential culprits for this deviation. One is that it appears the true scaling with number of rounds is not quite $r^4$, and that deviation becomes pronounced in the large round limit. In fact, were we instead to use two significant figures to estimate the logical error scaling with rounds, even a small change to $r^{3.8}$ would yield good agreement with the observed logical error rates. Another could be the slightly optimistic extrapolation of the gap distribution to many rounds observed in Fig. 7. However, since changing the inner code distance by even one significantly changes the logical error rate, we can obtain a fairly good overhead estimate from extrapolations that are within an order of magnitude. Furthermore, projecting overheads from scaling laws should provide conservative estimates, as the scaling laws tend to predict higher logical error rates.

### Footprint estimates

Having given evidence that these heuristic scaling approximations provide plausible, conservative estimates of the logical error rates, we use them to project the footprints required to hit different target logical error rates. These footprints include the overhead introduced by the workspace required to measure the checks, the overhead introduced by the yokes themselves, as well as the overhead of having an interstitial space between surface codes to mediate the lattice surgery. We include estimates for cold storage using both 1D and 2D

yoked surface codes, as well as hot storage using 1D yoked surface codes - see Fig. 9. Note that we report the target error per round, rather than per the $d$ rounds required to perform a primitive logical operation. Consequently, the teraquop regime begins around the $10^{-13} - 10^{-14}$ logical error rate mark in the figure.

In the teraquop regime, 1D yoked surface codes provide hot storage with nearly twice as many logical qubits per physical qubit as normal surface codes. 2D yoked surface codes provide cold storage with nearly three times as many logical qubits per physical qubit as normal surface codes. In both cases, as the target logical error rate decreases, the benefits of yoked surface codes become more pronounced. See also Supplementary Note 1 for overhead estimates using a standard uniform depolarizing error model, which is typically less noisy than SI1000 at the same parameterized noise rate[37]. In that model, reliably executing a trillion logical-qubit-rounds requires approximately 350 physical qubits per logical qubit, as opposed to 500 physical qubits per logical qubit in the above SI1000 model.

### Discussion

In this paper, we described how to yoke surface codes by measuring row and column logical parity checks along grids of surface codes. We estimate that yoked surface codes achieve nearly three times as many logical qubits per physical qubit as standard surface codes when operating in the teraquop regime at a physical error rate of $10^{-3}$. We also described a hierarchical architecture of hot and cold storage, where we can trade density of encoding for ease of memory access. In the context of hot storage, we estimate that yoked surface codes

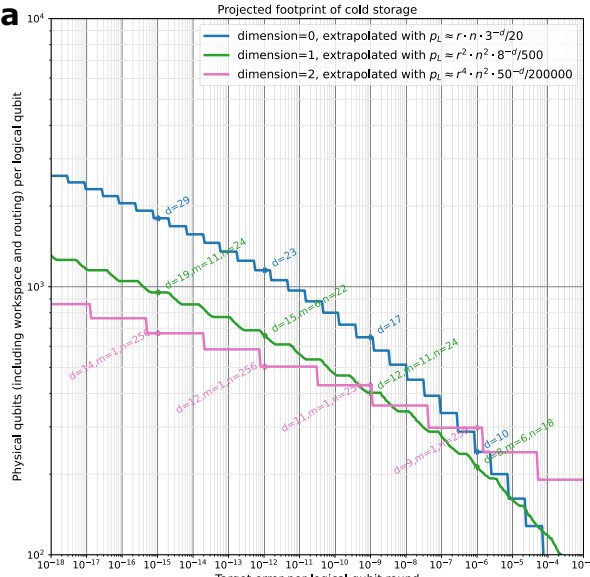

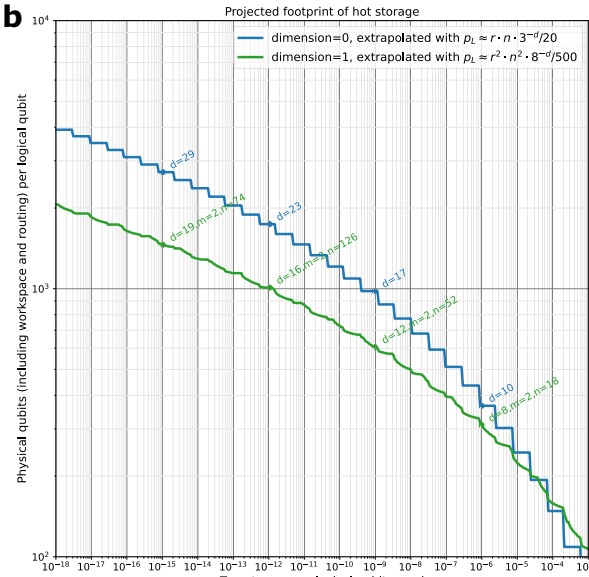

**Fig. 9 | Extrapolated footprints.** These include 0D (standard), 1D, and 2D yoked surface codes, and account for the workspace and access hallway overheads shown in Figs. 1 and 6. **a** Projections for cold storage and (**b**) projections for hot storage. For each target logical error rate, various patch diameters $d$, block sizes $n$, and number of blocks $m$ are tried. The most efficient layout that meets the target and encodes at most 250 logical qubits is identified using the scaling approximations. A patch of diameter $d$ is assumed to cover $2(d+1)^2$ physical qubits to leave some buffer space for lattice surgery. The yoked hot storage estimates target an access hallway utilization of 40%. Footprint estimates assume SI1000 noise at an error rate of $10^{-3}$. Standard circuit-level depolarizing noise footprint estimates can be found in Supplementary Note 1.

achieve nearly twice as many logical qubits per physical qubit as standard surface codes while keeping the logical qubits easily accessible during a computation. Reducing the quantity of qubits required by surface codes is extremely useful because this large quantity is perhaps the worst aspect of the surface code. While this is likely not as dramatic as reductions obtainable using LDPC codes with long-range connections, yoked surface codes do not require any additional connectivity.

Although we have focused on providing simple memory overhead estimates, there are several questions left unanswered. First, as we enter the large-scale error correction regime, it will be important to develop tooling that makes these types of simulations feasible. Currently, building the logical stim circuits is a hassle, and Monte Carlo simulations at this scale are difficult to perform exactly. Building automated tools out of the ZX calculus[38] would be helpful towards performing full-scale simulations of hierarchical memories. Ultimately, we only provide evidence of these savings through extrapolations - full-scale simulations are needed to verify the precise overhead saved. Due to the conservative choices of hook error suppression and scaling approximations, we expect that the overheads we report can be improved.

In identifying candidate outer codes to concatenate the surface code into, we focused mostly on coding rate. Choosing a code with a high rate is vital due to the qubit overhead introduced by the underlying surface codes. However, since any outer code can be generically laid out in 1D, we could look towards more complicated outer codes to try to further reduce the overhead of yoked surface codes.

Finally, it is important to understand hierarchical memories in the context of a fault-tolerant circuit. We have not carefully addressed questions of where and when these memories should be used in a computation, what sort of savings we can expect in that setting, how to gracefully encode, extract, and operate on encoded qubits, and so forth. These types of questions are important in assessing the overhead of applications of fault-tolerant quantum computers[3]. Ultimately, we believe that hierarchical memories like the one presented here hold promise for significantly reducing the cost of two-dimensional surface-code-based fault-tolerant quantum algorithms.

## Data availability

The stats collected and SketchUp renderings for this paper are available at[39].

## Code availability

The circuit generation and plotting code for this paper are also available at[39].

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

## Acknowledgements

We thank Jahan Claes, Austin Fowler, and Matt McEwen for helpful discussions, as well as Noah Shutty for writing the correlated minimum weight perfect matching decoder that we used (derived from PyMatching). We thank the Google Quantum AI team for creating an environment where this work was possible.

## Author contributions

Peter Brooks did groundwork on using complementary gaps for decoding. Cody Jones had the idea of concatenating a $Y^{\otimes n}$ stabilizer over the surface code to double the code distance. Craig Gidney got excited by Cody's idea, did simulations with circuit noise on the inner code and code capacity noise on the outer code, extended the idea to 1D parity check codes, wrote the first draft of the paper, and constructed the 2D parity check codes for an appendix. Michael Newman got excited by Craig's results, extended the simulations to include gapped phenomenological noise, compiled some of the topological circuits, pushed to focus more on the 2D parity check codes, and rewrote the paper.

## Competing interests
The authors declare no competing interests.
