## [Transparent Peer Review file · Nature Communications]

Yoked surface codes

Corresponding Author: Dr Michael Newman

Version 0:

Reviewer comments:

Reviewer #1

(Remarks to the Author)

This work presents a concatenated architecture for reducing the overhead in fault-tolerant quantum computation. The main thesis of this paper is to use the well-studied structure of the surface code and the ability to perform joint logical measurements across multiple patches using lattice surgery in order to measure the large-weight stabilizers of the proposed parity check (or array codes). I will use the term array code in order to avoid confusion with the general use of the term parity check in codes. These array codes are desirable to their high encoding rate, albeit they have rather low distance. That being said, the distance benefits are significant enough to reduce the overall overhead of a full architecture as opposed to just taking the naive perspective of many copies of simple surface codes at a given distance. I think the main conclusion of this paper is quite interesting, that simple concatenation with structure can yield interesting reductions in overhead when the underlying structure is sufficiently simple.

The main technical contribution of the paper I would argue is the analysis and subsequent numerical simulations of concatenated code. Due to the large codeblock size that the authors' aimed to study, as well as the needed time overhead for fault tolerance, simulating a full stack of a computation with a given noise model quickly becomes computationally expensive. As such, the authors argue for a hierarchical noise structure using a technique called complimentary gaps. The specifics of this seem quite new and novel. To my understanding, the study of these gaps is similar to a form of soft decoding in that when a given syndrome is measured on a small patch of surface code, it tells the higher level decoder how confident one is in the decision that was reached. This in turn is turned into a method for weighing the edge weights of the decoder on the outer level when trying to run min-weight perfect matching. The authors are admittedly rather conservative in some of the bounds they make in this section thus their numerics are believable and are probably on the pessimistic side.

In addition, the authors provide a long discussion on estimating the overhead based on the logical error rates inferred from their numerics. This shows a clear separation in overhead for the concatenated dimension-1 and -2 codes as opposed to that of the un-concatenated dimension-0 code, where this improvement gets further emphasized the lower (larger distance) the logical error rate needs to be.

A few questions:

In Section 5B, the authors infer from numerics the effective logical distances of their concatenated scheme. As they state, these fall short from the distances expected in the respective concatenated schemes. Is this only due to the pessimistic/conservative approach to the approximations they make in the numerics? It would seem that this should not be the result of a min-weight decoder, but perhaps some approximations made here yield lower-weight errors? As a follow up, if do to some approximations in the numerics/noise model for the concatenated model, do the authors expect that implementing a full min-weight decoder on real hardware will be too punitive and similar approximations would have to be made? If so, what approaches towards efficiency to the authors envision without sacrificing the desired logical accuracy?

In the conclusion, the authors have the following sentence: "While this is likely not as dramatic as reductions obtainable using LDPC codes with long-range connections, it does not require any additional connectivity and is fairly simple to lay out." Did the authors spend any time pursuing other high rate QEC codes to concatenate with the surface code? The natural candidate would be the class of hypergraph product codes, which can have distance scaling like L , where L is the outer code length at the cost of L^2 blocks and encoding a constant rate of logical qubits. It seems from the construction, that the outer stabilizers measuring in the array codes are as difficult, if not more so, than those of the HGP given that they are larger weight. Moreover, the HGP can be laid out in the same 2d architecture where stabilizer checks are measured along rows

and columns. I suppose the main drawback is the difficulty in parallelizing these measurements with the layout prescribed and that it may require additional “hallway” space, yet would there be other unforeseen drawbacks?

(Remarks on code availability)

Reviewer #2

(Remarks to the Author)

In this manuscript, the authors present a concatenation protocol of the surface code with a high density parity check outer code. The authors present some families of outer codes which are suitable for concatenation with the surface code and show how the parity checks of the outer code can be measured using lattice surgery with the surface code. The authors also present convincing numerical evidence that for memory, fewer qubits would be required in such a concatenation protocol compared to using bare surface codes.

Overall, concatenating the surface code with an outer code, whether it be a high density parity check code or some other topological code is a very interesting idea. Whether it presents a more suitable architecture for performing computation remains to be seen, but this paper certainly paves the way for further exploration. I enjoyed reading this paper and recommend it for publication. However I do have some comments that I believe could improve the paper prior to being published.

COMMENTS:

- 1) The authors use many figures showing the lattice surgery pipeline for performing the necessary Pauli measurements. However I believe that figures 2,4 and 5 may be a bit abrupt of a presentation. To make the paper slightly more self-contained, I believe a figure similar to figure 3 could be used to show an example of a Mzzzzz measurement. That is, showing the transformations of the surface code patches through time. Otherwise, upon the first few readings, it is quite hard to follow.
- 2) I think the authors should provide a more in depth explanation as to why timeline failures result in correlated errors when measuring the parity checks of the outer code. In particular, a lattice surgery protocol can have both space like and timeline failures, and showing in more detail how both failure mechanisms translate to errors when measuring parity checks of the outer code could be beneficial.
- 3) Given the hallway access chosen by the authors, surface code patches need to be rotated in order to have access to both X and Z type boundaries. It wasn't clear to me that an architecture with larger hallway access, but where both boundaries of the surface code patches can be accessed, is more costly. Can the authors add a more thorough explanation as to why?
- 4) In section 4, it could help to show a figure of the modified error graph when performing MWPM conditioned on the complementary logical outcome. Again on a first reading, I found it somewhat confusing as to how the edges of the graph for the outer codes were re-weighted based on the MWPM outcomes of the inner surface codes. There is a type of pre-conditioning matching algorithm being implemented here, and I believe a better explanation is needed, perhaps with an example.

Typo:

- Page 6, “boundary. [36]” —> boundary [36].”

(Remarks on code availability)

Version 1:

Reviewer comments:

Reviewer #1

(Remarks to the Author)

The authors have adequately addressed the comments and questions from the reviewers in their rebuttal and there are no further outstanding questions. I recommend publication of this work.

(Remarks on code availability)

Reviewer #2

(Remarks to the Author)

The authors have addressed all of my concerns. I recommend the article for publication.

(Remarks on code availability)

REVIEWER COMMENTS

Reviewer #1 (Remarks to the Author):

This work presents a concatenated architecture for reducing the overhead in fault-tolerant quantum computation. The main thesis of this paper is to use the well-studied structure of the surface code and the ability to perform joint logical measurements across multiple patches using lattice surgery in order to measure the large-weight stabilizers of the proposed parity check (or array codes). I will use the term array code in order to avoid confusion with the general use of the term parity check in codes. These array codes are desirable to their high encoding rate, albeit they have rather low distance. That being said, the distance benefits are significant enough to reduce the overall overhead of a full architecture as opposed to just taking the naive perspective of many copies of simple surface codes at a given distance. I think the main conclusion of this paper is quite interesting, that simple concatenation with structure can yield interesting reductions in overhead when the underlying structure is sufficiently simple.

The main technical contribution of the paper I would argue is the analysis and subsequent numerical simulations of concatenated code. Due to the large codeblock size that the authors' aimed to study, as well as the needed time overhead for fault tolerance, simulating a full stack of a computation with a given noise model quickly becomes computationally expensive. As such, the authors argue for a hierarchical noise structure using a technique called complimentary gaps. The specifics of this seem quite new and novel. To my understanding, the study of these gaps is similar to a form of soft decoding in that when a given syndrome is measured on a small patch of surface code, it tells the higher level decoder how confident one is in the decision that was reached. This in turn is turned into a method for weighing the edge weights of the decoder on the outer level when trying to run min-weight perfect matching. The authors are admittedly rather conservative in some of the bounds they make in this section thus their numerics are believable and are probably on the pessimistic side.

In addition, the authors provide a long discussion on estimating the overhead based on the logical error rates inferred from their numerics. This shows a clear separation in overhead for the concatenated dimension-1 and -2 codes as opposed to that of the un-concatenated dimension-0 code, where this improvement gets further emphasized the lower (larger distance) the logical error rate needs to be.

We thank reviewer #1 for their kind words and their attention to the multiple novel elements of the paper, include the use of complementary gaps. We do indeed believe that the use of these gaps will become important in both simulation and in the context of different fault-tolerant protocols (beyond just its deployment in the concatenated scheme presented here).

A few questions:

In Section 5B, the authors infer from numerics the effective logical distances of their concatenated scheme. As they state, these fall short from the distances expected in the

respective concatenated schemes. Is this only due to the pessimistic/conservative approach to the approximations they make in the numerics? It would seem that this should not be the result of a min-weight decoder, but perhaps some approximations made here yield lower-weight errors?

This is a great question from the reviewer. To summarize the status here, the effective distance comes into play with respect to an empirically computed error suppression factor λ , which represents the amount of error suppression we see when incrementing the code distance by one.

The error suppression factor differs somewhat from what one might expect when extrapolating from the distance increase, although it is relatively close. Using the 'noisier' SI1000 model from the main text, for the usual "unyoked" surface codes, this factor is approximately $\lambda=3$. With 1D yoked surface codes, we see a factor of $\lambda=8$ (implying an "effective" distance increase of $\log_3(8) \approx 1.9x$ rather than $2x$) and for 2D yoked surface codes, we see a factor of $\lambda=50$ (implying an "effective" distance increase of $\log_3(50) \approx 3.6x$ rather than $4x$).

However, we don't believe that this is the result of lower-weight errors causing failures (indeed, we expect that the distance of the code is preserved). One hypothesis is that our minimum-weight path approximations are breaking down slightly. In particular, it can be the case that paths of slightly longer length are contributing to the error rate, so we don't see the error rate being suppressed quite as quickly as would be predicted from the distance increasing.

As a follow up, if do to some approximations in the numerics/noise model for the concatenated model, do the authors expect that implementing a full min-weight decoder on real hardware will be too punitive and similar approximations would have to be made? If so, what approaches towards efficiency to the authors envision without sacrificing the desired logical accuracy?

Apologies if we misunderstand the question, which we interpret as "would implementing a full minimum-weight perfect matching decoder" in real hardware on such a system be feasible? This could stand as an alternative to computing complementary gaps, so long as the yoke observables are designed to run along boundaries (so that the decoding problem, with the inclusion of the yoke detectors, remains matchable). We believe that both approaches are reasonable. One negative aspect of designing a monolithic matching problem is that one might have to compile the yoke checks to ensure this "observable along the boundary" property, which could increase the spacetime overhead of the check. One negative aspect of explicitly computing complementary gaps is that it might be slower than the usual decoding problem, although there has been some very recent progress on ways to speed this up in e.g. [arXiv:2405.07433](https://arxiv.org/abs/2405.07433).

In the conclusion, the authors have the following sentence: "While this is likely not as dramatic as reductions obtainable using LDPC codes with long-range connections, it does not require any additional connectivity and is fairly simple to lay out." Did the authors spend any time pursuing other high rate QEC codes to concatenate with the surface code? The natural

candidate would be the class of hypergraph product codes, which can have distance scaling like L , where L is the outer code length at the cost of L^2 blocks and encoding a constant rate of logical qubits. It seems from the construction, that the outer stabilizers measuring in the array codes are as difficult, if not more so, than those of the HGP given that they are larger weight. Moreover, the HGP can be laid out in the same 2d architecture where stabilizer checks are measured along rows and columns. I suppose the main drawback is the difficulty in parallelizing these measurements with the layout prescribed and that it may require additional “hallway” space, yet would there be other unforeseen drawbacks?

Trying out other codes as the outer code to concatenate into is an interesting research direction, and indeed, looking at other LDPC codes would be one avenue. However, we do think that choosing codes that have very high rates are especially desirable simply because surface codes are themselves so expensive.

For example, in the supplement we look at “standard” circuit-level depolarizing noise, and find that ~ 300 physical qubits/logical qubit can suppress down to 10^{-12} error rates. We can perform a back-of-the-envelope comparison to see what choosing a code with a lower rate might achieve. We assume that each surface code takes of $2 \cdot (d+1)^2$ qubits (to give some buffer space for lattice surgery). That means that even using distance-5 surface codes as inner codes would necessitate using a code family whose rate was at least $300/72 = 0.24$, which is quite high for most smaller LDPC codes that we know of (since a small code block size is also desirable). Furthermore, distance-5 surface codes are quite small, and as the reviewer points out, we may need them to preserve information for a long period of time depending on the complexity of the outer LDPC checks.

The benefit, as the reviewer also points out, is that the outer codes would have higher distance. Indeed, our expectation is that for sufficiently low target error rates, such code families could be more efficient for this reason. However, we are concerned with specific target logical error rates relevant to the expected spacetime requirements for fault-tolerant quantum algorithms. In this regime, when concatenating with the surface code, it seems like having a very high rate is a priority. However, we certainly agree that careful studies should be done to look into other promising outer code families, HGP codes being a natural first place to look. We have added a short paragraph to the conclusion addressing this possibility: “In identifying candidate outer codes to concatenate the surface code into, we focused mostly on coding rate. Choosing a code with a high rate is vital due to the qubit overhead introduced by the underlying surface codes. However, since any outer code can be generically laid out in 1D, we could look towards more complicated outer codes to try to further reduce the overhead of yoked surface codes.”

Reviewer #2 (Remarks to the Author):

In this manuscript, the authors present a concatenation protocol of the surface code with a high density parity check outer code. The authors present some families of outer codes which are suitable for concatenation with the surface code and show how the parity checks of the outer code can be measured using lattice surgery with the surface code. The authors also present convincing numerical evidence that for memory, fewer qubits would be required in such a concatenation protocol compared to using bare surface codes.

Overall, concatenating the surface code with an outer code, whether it be a high density parity check code or some other topological code is a very interesting idea. Whether it presents a more suitable architecture for performing computation remains to be seen, but this paper certainly paves the way for further exploration. I enjoyed reading this paper and recommend it for publication. However I do have some comments that I believe could improve the paper prior to being published.

COMMENTS:

1) The authors use many figures showing the lattice surgery pipeline for performing the necessary Pauli measurements. However I believe that figures 2,4 and 5 may be a bit abrupt of a presentation. To make the paper slightly more self-contained, I believe a figure similar to figure 3 could be used to show an example of a Mzzzzzz measurement. That is, showing the transformations of the surface code patches through time. Otherwise, upon the first few readings, it is quite hard to follow.

This is a great suggestion as many in the field aren't necessarily familiar with topological diagrams or the ZX-calculus. We have added the suggested diagram to Figure 2 in order to clarify the connection between topological diagrams and the stabilizer measurements that comprise the corresponding lattice surgery steps.

2) I think the authors should provide a more in depth explanation as to why timeline failures result in correlated errors when measuring the parity checks of the outer code. In particular, a lattice surgery protocol can have both space like and timeline failures, and showing in more detail how both failure mechanisms translate to errors when measuring parity checks of the outer code could be beneficial.

We have modified Figure 3 to include both spacelike and timelike errors, which on the left hand side correspond to hook and measurement errors, but whose roles are reversed in the right hand lattice surgery construction. We have emphasized the role of the measurement error by including a correlation surface that identifies the measurement error with a data error immediately before and after the yoke measurement. Hopefully this helps clarify both kinds of errors that come up during the lattice surgery measurements.

3) Given the hallway access chosen by the authors, surface code patches need to be rotated in order to have access to both X and Z type boundaries. It wasn't clear to me that an architecture with larger hallway access, but where both boundaries of the surface code patches can be accessed, is more costly. Can the authors add a more thorough explanation as to why?

This is a good question. In the context of hot storage, access hallways are the means by which we measure the yoke checks. We could indeed expose both types of boundaries of the surface code. The potential benefits are three-fold. First, it would shorten the syndrome extraction cycle (as the patch rotations would no longer be necessary), which would also lead to a lower error rate. Second, it simplifies performing computations, as we would no longer have to account for patch rotations to expose the necessary patch at the correct time. Finally, having a layout with both bases exposed could facilitate used 2D yoked surface codes, depending on the layout.

It is possible that some such layout could prove beneficial. However, the cost of the most naive layout (by allowing access hallways of rows and columns) would increase the spatial footprint by at least 1.5x, which is quite expensive. Simply put, in back-of-the-envelope simulations, the benefits of shorter cycles or even using 2D yoked surface codes do not outweigh this cost. Furthermore, since they are not very large, we do not expect the patch rotations to be overly burdensome when performing operations. However, determining this cost will be important when estimating savings in the context of a particular fault-tolerant computation.

We have added this explanation to the Lattice Surgery Constructions section: "In fact, storage could be even hotter than we consider here, by having each surface code patch expose two boundaries ~\cite{litinski2019gameofsurfacecodes,litinskyhypercubeshor2023}. However, we expect the additional space cost (naively at least 1.5x) of exposing two boundaries would likely mitigate any benefit of using 2D yoked surface codes or the computational benefit of having both bases immediately available. Consequently, qubits in hot storage are rotated on an as-needed basis when accessing different boundaries. The need to perform these rotations is a key consideration when laying out an algorithm."

4) In section 4, it could help to show a figure of the modified error graph when performing MWPM conditioned on the complementary logical outcome. Again on a first reading, I found it somewhat confusing as to how the edges of the graph for the outer codes were re-weighted based on the MWPM outcomes of the inner surface codes. There is a type of pre-conditioning matching algorithm being implemented here, and I believe an better explanation is needed, perhaps with an example.

We have expanded on the explanation in section 4 - hopefully this helps clarify the reweighting procedure. Indeed, the reviewer is correct that it is a "pre-conditioning" matching algorithm where the prior (i.e. edge weights) changes based on the detection events observed in the inner surface codes. We added the following: "This information is extremely helpful and can be used in decoding the outer code. 1D and 2D parity check codes can themselves be decoded using minimum-weight perfect matching, and so we use these complementary gaps as edge weights in the outer error graph. These edge weights represent the cost of flipping a minimum-weight

matching to a complementary matching. The complementary gap is converted to a probability which will serve as the effective error rate of the edge of the outer error graph it corresponds to. Note that in practice one must XOR the syndrome produced by the minimum-weight perfect matching into the observed syndrome, as choosing the outer edge represents the outer decoder selecting the complementary matching in place of the minimum-weight matching. Relative to the usual decoding problem, the outer decoder's prior changes shot-by-shot depending on the configuration of detection events in the inner codes. While we have selected matchable outer codes, this procedure easily generalizes to setting the prior of any outer decoding problem (possibly replacing edge probabilities with hyperedge probabilities) so long as one has a suitable outer code decoder."

Typo:

- Page 6, "boundary. [36]" —> boundary [36]."

Fixed, thank you.